health and disease and epidemiology/ mathematical modelling

contact tracing, COVID-19, non-pharmaceutical interventions, renewal equation, time since infection, resource constraints

**Author for correspondence:**
Jianhong Wu
e-mail: wujh@yorku.ca

# A renewal equation model to assess roles and limitations of contact tracing for disease outbreak control

Francesca Scarabel[1,2,3], Lorenzo Pellis[4,5], Nicholas H. Ogden[6] and Jianhong Wu[1,2]

[1]LIAM—Laboratory for Industrial and Applied Mathematics, Department of Mathematics and Statistics, and [2]Fields-CQAM Laboratory of Mathematics for Public Health, York University, Toronto, Ontario, Canada
[3]CDLab—Computational Dynamics Laboratory, Department of Mathematics, Computer Science and Physics, University of Udine, Italy
[4]Department of Mathematics, University of Manchester, UK
[5]The Alan Turing Institute, London, UK
[6]Public Health Risk Sciences Division, National Microbiology Laboratory, Public Health Agency of Canada, St-Hyacinthe, Quebec, Canada

 FS, 0000-0003-0250-4555; LP, 0000-0002-3436-6487; NHO, 0000-0002-1062-7283; JW, 0000-0003-0052-5336

We propose a deterministic model capturing essential features of contact tracing as part of public health non-pharmaceutical interventions to mitigate an outbreak of an infectious disease. By incorporating a mechanistic formulation of the processes at the individual level, we obtain an integral equation (delayed in calendar time and advanced in time since infection) for the probability that an infected individual is detected and isolated at any point in time. This is then coupled with a renewal equation for the total incidence to form a closed system describing the transmission dynamics involving contact tracing. We define and calculate basic and effective reproduction numbers in terms of pathogen characteristics and contact tracing implementation constraints. When applied to the case of SARS-CoV-2, our results show that only combinations of diagnosis of symptomatic infections and contact tracing that are almost perfect in terms of speed and coverage can attain control, unless additional measures to reduce overall community transmission are in place. Under constraints on the testing or tracing capacity, a temporary interruption of contact tracing may, depending on the overall growth rate and prevalence of the infection, lead to an irreversible loss of control even when the epidemic was previously contained.

# 1. Introduction

Used in combination with other public health efforts for diagnosis of symptomatic individuals or preventive screening, contact tracing has the potential to detect asymptomatic or pre-symptomatic individuals rapidly and efficiently, thus preventing further transmission chains. Moreover, by targeting specifically individuals who have been exposed to the infection, it allows resources to concentrate on the population at risk, in contrast with other untargeted interventions like mass testing, physical distancing or mass quarantining. For these reasons, contact tracing is one of the most cost-effective and widely adopted non-pharmaceutical interventions to counteract the spread of infectious diseases in the absence of effective treatments and vaccines. It contributed successfully, for example, to control the spread of coronaviruses like severe acute respiratory syndrome (SARS) and Middle East respiratory syndrome (MERS) [1], and the Ebola virus [2–4]. In the ongoing battle against the severe respiratory disease caused by the coronavirus SARS-CoV-2, contact tracing has been used to mitigate the global spread of COVID-19 almost universally but with various degrees of success and failure [5].

Many studies, based on different mathematical models, investigate mechanisms of contact tracing and its efficacy. Modelling contact tracing is challenging: a satisfactory mathematical description should account at least for concurrent screening/diagnosis programmes that can initiate contact tracing, and the contacts between individuals that occurred prior to that moment. For these reasons, many mathematical models have been formulated as stochastic branching processes [6–8] or agent-based models [9–13], which allow to keep track of the epidemiological status of each individual in the population together with all their infectious contacts. As such, these models require large computations to simulate the epidemics and, although they can provide evidence of the simulation results, extrapolating general insights can be challenging. The few deterministic models available in the literature [6,14–16] are typically obtained as approximations of a corresponding stochastic model under simplifying assumptions. Here, we present what, to our knowledge, is the first deterministic model that overcomes these limitations by employing both delayed (in time) and advanced (in time since infection) temporal variables, as explained below, to describe the full nonlinear epidemic dynamics in real time. We note that similar arguments, which also lead to advanced time-since-infection terms, have been used in [6–8] to construct a discrete-generation stochastic branching process, which, however, is limited to the early (linear) epidemic phase and does not describe the real-time epidemic dynamics.

We develop a deterministic mixed-type renewal equation model for transmission dynamics that incorporates diagnosis of individuals from symptoms and contact tracing. We focus on *forward* contact tracing [6,17]: when an infected individual is detected, only its suspected secondary cases are traced. By contrast, *backward* contact tracing aims at finding the infector of the index case [18]. The modelling framework is motivated by a classical renewal equation approach [19,20] where rates are described as a function of time, or *age*, since infection (i.e. the time passed since the infection event), thus allowing to incorporate realistic distributions of infectiousness and incubation period.

The contact tracing process is modelled by describing the probability that an infected index case is detected by contact tracing. This event happens only if the infector of the index case was detected, by diagnosis or contact tracing, hence triggering the search of the secondary cases. This formulation naturally leads to a mixed-type integral equation, delayed in time and advanced in the age since infection, since it depends both on the infection event in the past and on the likelihood that the infector is later tested and traced: in terms of age since infection, the infector is always 'older' than the infectee. By describing the relation between infectee and infector, the model enables us to link the probability of being traced with the overall epidemic in the population and to capture the intrinsic correlation between diagnosis of individuals and contact tracing, which is typically one of the main bottlenecks of deterministic approaches [16].

For an emerging outbreak, we provide explicit formulas for the basic reproduction number $R_0$, describing the average number of secondary cases generated by one infectious individual in an otherwise susceptible population, and the effective reproduction numbers in the presence of isolation of individuals upon diagnosis and contact tracing. We show that the outbreak can be brought under control only if the proportion of transmission that occurs before isolation is less than $1/R_0$, regardless of how isolation is achieved. This is in agreement with the observation by Fraser *et al.* [16], that relate the controllability of an epidemic with $R_0$ and the fraction of pre-symptomatic transmission. This also shows that, for a disease that is highly infectious or with large amount of pre-symptomatic transmission, a contact tracing programme is unlikely to be effective unless other social distancing measures are in place.

The public health response to an emerging outbreak in the absence of vaccine involves a coordination and combination of measures including diagnostic testing, contact tracing and reduction of community

transmission via personal protective equipment, physical distancing or, in the case of COVID-19, mass quarantine. Some measures are more disruptive than others to private life and society, and each one impacts the infection dynamics differently. Estimating the feasibility and cost-effectiveness of each measure is key for a successful containment, and requires careful evaluation of several epidemiological indicators.

We here ignore the effect of delays in contact tracing and focus on two different characteristics: the tracing *window*, defined as the length of the period of time preceding detection during which contacts are traced and isolated, and the tracing *coverage*, defined as the fraction of contacts that are effectively traced. We use parameters from COVID-19 to illustrate several mechanisms of success and failure of contact tracing to prevent the outbreak, in relation to diagnosis of symptomatic individuals (in terms of speed and efficacy of diagnosis) and additional public health measures that aim at reducing overall transmission. Finally, we address a question of public health relevance in the presence of limited testing or tracing capacity, namely the short-term interruption of contact tracing. We show that, with limitations on capacity, the loss of control during a short-term interruption of contact tracing may not be reversible, in the sense that reintroducing contact tracing may not recover the negative growth rate pre-interruption, if capacity is breached. This example underscores how, in the presence of resource constraints, even measures that may seem of minor impact assume an important role in maintaining control [1,21].

The mathematical model is introduced in §2 in the fully nonlinear context, whereas important concepts in the case of an emerging outbreak, including the effective reproduction numbers in the presence of diagnosis and contact tracing, are the focus of §3. Finally in §4, we consider some specific scenarios and focus on some epidemiological and public health insights from our model-based simulations and analyses.

# 2. Model formulation

We assume that individuals can either be detected because they develop symptoms (or from any other screening programme that does not depend on the overall epidemic) or from contact tracing. We talk about *diagnosis* of individuals if they are detected from symptoms and *tracing* if they are detected via a contact tracing process, whereas we talk about *detection* of individuals either from symptoms or contact tracing. We only consider *forward* contact tracing: when one individual is detected (index case) and contact tracing is initiated, only secondary contacts of the index case are isolated, rather than their infector. We assume that, upon detection (whether it is by diagnosis or contact tracing), individuals are immediately isolated and do not contribute to the force of infection, and contact tracing of their secondary cases is immediately initiated. This also implies that contact tracing is iterative, in the sense that contacts of contacts are indefinitely (and instantaneously) traced. Throughout this study, we assume that only infected individuals can trigger contact tracing. In reality, this would rely on a monitoring protocol for which all contacts are immediately tested, so that it is immediately (and with certainty) known which individuals are actually infected and trigger further tracing only for those.

## 2.1. The diagnosis process

We denote by $h_d(\tau)$ the rate of an individual with age since infection (hereafter, ASI) $\tau$ being diagnosed for symptoms. We will assume that this is a given known model ingredient, determined both by the disease and by the surveillance strategy, and, for simplicity, we assume it is independent of time (although, in general, diagnosis and testing protocols may vary during the course of an epidemic). Note that, in prescribing $h_d$, we are implicitly assuming that distinct individuals are independent of each other in terms of diagnosis. Typically, $h_d$ has compact support (i.e. $h_d(\tau) = 0$ if $\tau$ is larger than a given value), since infected individuals can be diagnosed only within a certain time after infection (for example, because after a while the diagnostic test gives a negative result). Let $\mathcal{F}_d(\tau)$ denote the probability that an infected individual has not yet been diagnosed by ASI $\tau$. This is given by

$$\mathcal{F}_d(\tau) = e^{-\int_0^\tau h_d(\sigma)d\sigma} = 1 - F_d(\tau),\tag{2.1}$$

where $F_d(\tau)$ denotes the cumulative distribution function of the time to diagnosis, i.e. the probability that an individual has been diagnosed before ASI $\tau$. From (2.1), it is also clear that $h_d$ can be computed from the prescribed cumulative distribution by

$$h_d(\tau) = -\frac{d}{d\tau}\log(1 - F_d(\tau)).\tag{2.2}$$

Assume $\lim_{\tau \to \infty} F_d(\tau) = \varepsilon_d$, with $\varepsilon_d \in [0, 1]$. Then $\varepsilon_d$ represents the fraction of individuals that are eventually diagnosed: if $\varepsilon_d = 1$, then every infected individual will eventually develop symptoms and be diagnosed; if $\varepsilon_d < 1$, a fraction $1 - \varepsilon_d$ of infected individuals will escape diagnosis. Note that the probability $\mathcal{F}_d$ of not being diagnosed at time $\tau$ after infection can be related to the distribution of the incubation time by specifying a suitable density function for the delay between symptom onset and diagnosis, as suggested in [16].

## 2.2. The contact tracing process

Let $h_c(t, \tau)$ denote the rate at which an individual of ASI $\tau$ at time $t$ is detected by contact tracing. The probability that an individual is traced in $[t, t + dt]$ is given by $h_c(t, \tau)\,dt$, and the probability that an individual of ASI $\tau$ at time $t$ has not yet been detected by contact tracing is

$$\mathcal{F}_c(t, \tau) = e^{-\int_0^\tau h_c(t-\tau+\sigma,\sigma)d\sigma}, \quad t, \tau > 0. \tag{2.3}$$

Note that we explicitly include the dependence of $h_c$ on time $t$, since the contact tracing process is intrinsically connected with the underlying epidemic. Moreover, since the contact tracing process alters, but in turn is also affected by, the transmission process, $h_c(t, \tau)$ is an unknown dynamical quantity in this model.

An individual can be traced only if their infector (or any earlier infector along a chain of transmission that is iteratively traced) is diagnosed. Hence, $\mathcal{F}_c$ depends on the diagnosis, and hence the ASI, of such an infector. On the other hand, the probability $\mathcal{F}_d$ of an individual not being diagnosed depends only on their own ASI, and it is therefore independent of the diagnosis of other individuals in the population and, in particular, of $\mathcal{F}_c$. Hence the probability that an individual has not been isolated (either by diagnosis or contact tracing) by ASI $\tau$ is simply given by the product

$$\mathcal{F}(t, \tau) := \mathcal{F}_d(\tau)\mathcal{F}_c(t, \tau). \tag{2.4}$$

## 2.3. A renewal equation for the incidence

We first define an equation for the incidence following a classical renewal equation approach [19,20]. Denoting by $y(t)$ the incidence at time $t$, we can write

$$y(t) = \frac{S(t)}{N_0} \Lambda(t), \tag{2.5}$$

where $N_0$ is the total population (assumed to be constant), and $S(t)$ and $\Lambda(t)$ are, respectively, the total number of susceptibles and the total force of infection in the population at time $t$. The susceptible population satisfies $S'(t) = -y(t)$, $S(0) = N_0$, hence

$$S(t) = N_0 - \int_0^t y(s)\,ds. \tag{2.6}$$

Let $\beta(\tau)$ describe the infectiousness, that is, the per capita contribution to the force of infection, of an infected individual with ASI $\tau$. As for $h_d$, in prescribing $\beta$ we are implicitly assuming that distinct individuals are independent of each other in terms of transmission. We can also split $\beta(\tau) = \phi(\tau)\mathcal{F}_r(\tau)$, where $\mathcal{F}_r(\tau)$ describes the probability that an infected individual has not yet recovered at ASI $\tau$, and $\phi$ represents the infectivity of an individual that has not yet recovered. With this notation, the density of infective (not yet recovered) individuals at time $t$ with ASI $\tau$ is

$$I(t, \tau) = y(t - \tau)\mathcal{F}_r(\tau). \tag{2.7}$$

Assuming that detected individuals are instantly isolated and do not contribute further to the force of infection, the density of active infective individuals with ASI $\tau$, i.e. individuals who are infective and not isolated, at time $t$ is $y(t - \tau)\mathcal{F}_r(\tau)\mathcal{F}(t, \tau)$, with $\mathcal{F}$ defined in (2.4), and the total force of infection is

$$\Lambda(t) = \int_0^\infty \beta(s)y(t - s)\mathcal{F}(t, s)\,ds. \tag{2.8}$$

From (2.5) and (2.8), we obtain

$$y(t) = \frac{S(t)}{N_0} \int_0^\infty \beta(s)y(t - s)\mathcal{F}(t, s)\,ds, \tag{2.9}$$

which, for given $\mathcal{F}$, is a renewal equation for the incidence $y(t)$.

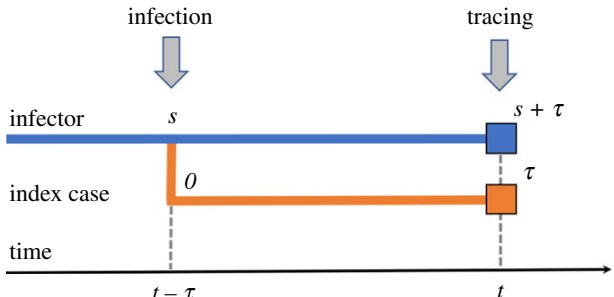

**Figure 1.** Scheme of the infection and (forward) contact tracing process. An index case with ASI $\tau$ at time $t$ is traced at time $t$ only if the infector is detected at time $t$. In the figure, infection happens at time $t - \tau$ when the infector has ASI $s$; the infector is then detected at time $t$, with ASI $s + \tau$, triggering the contact tracing.

## 2.4. An integral equation for the contact tracing rate

To describe the evolution of the contact tracing rate $h_c$ in time, we look at an index case that has ASI $\tau$ at time $t$ (i.e. was infected at time $t - \tau$), and compute the probability that the index case is detected by contact tracing in $[t, t + dt]$, which itself depends on the probability that contact tracing is initiated for the infector of the index case, see also figure 1.

We first study the relation between the ASI of infectee and infector. The probability $\psi(t, s)\, ds$ that a random infector at time $t$ has ASI in $[s, s + ds]$ is given by

$$\psi(t, s)\, ds = \frac{\beta(s)y(t - s)\mathcal{F}(t, s)\, ds}{\Lambda(t)} = \frac{S(t)}{N_0 y(t)} \beta(s)y(t - s)\mathcal{F}(t, s)\, ds. \tag{2.10}$$

Let $\varepsilon_c$ denote the tracing coverage (i.e. the fraction of contacts that are effectively traced). We can compute explicitly the probability $h_c(t, \tau)\, dt$ that an individual that was infected at time $t - \tau$ is traced in $[t, t + dt]$ (see also figure 1) by

$$h_c(t, \tau)\, dt = \varepsilon_c \int_0^\infty P(\text{infector has ASI in } [s, s + ds] \text{ at time } t - \tau)$$

$$\times P(\text{contact tracing initiated in } [t, t + dt] \mid \text{infector not detected before } t - \tau)$$

$$= \varepsilon_c \int_0^\infty (\psi(t - \tau, s)\, ds) \frac{\mathcal{F}(t, \tau + s)(h_d(\tau + s) + h_c(t, \tau + s))\, dt}{\mathcal{F}(t - \tau, s)}.$$

Note that the integral at the right-hand side is computed with respect to $s$. Simplifying $dt$ from both sides and using (2.10), we obtain

$$h_c(t, \tau) = \frac{\varepsilon_c S(t - \tau)}{N_0 y(t - \tau)} \int_0^\infty \beta(s)y(t - \tau - s)\, \mathcal{F}(t, \tau + s)(h_d(\tau + s) + h_c(t, \tau + s))\, ds, \tag{2.11}$$

where $\mathcal{F}$ is defined by (2.3) and (2.4), and $S$ is given by (2.6). Equation (2.11) is delayed in $t$ and advanced in $\tau$. In general, we assume that only the contacts that took place in a certain time window before detection of an individual are traced. Under this assumption, $h_c(t, \tau) = 0$ if $\tau$ is larger than the length of the tracing window. Assuming that $h_d$ is prescribed, we obtain a closed system by coupling (2.9) with (2.11). Assuming that the integration bounds are finite, which can be obtained by setting $\beta$ to zero after a certain time, the model can be solved numerically by a finite difference scheme using quadrature rules. The simulations in the next section were performed in MATLAB R2019a.

## 2.5. Interpretation of the model

The model is formulated in terms of the quantity $h_c$, which is by itself quite abstract and difficult to measure in practice. However, more concrete quantities of interest can be explicitly computed from $h_c$ and from the incidence $y$. First, given the incidence we can compute explicitly the number of susceptible individuals from (2.6), and the number of infected individuals from (2.7). Moreover, regarding the diagnosed and traced individuals we have

$$(\text{infected}) \text{ individuals diagnosed in } [t, t + dt] = dt \int_0^\infty y(t - s)\mathcal{F}(t, s)h_d(s)\, ds, \tag{2.12}$$

and

$$\text{(infected) individuals traced in } [t, t + dt] = dt \int_0^\infty y(t - s)\mathcal{F}(t, s)h_c(t, s)\,ds. \tag{2.13}$$

Constraints on the capacity of contact tracing resources can be modelled by imposing that the total number of contacts traced in a certain time interval $[t_1, t_2]$ is bounded

$$\int_{t_1}^{t_2} \int_0^\infty y(t - s)\mathcal{F}(t, s)h_c(t, s)\,ds\,dt < K,$$

where $K$ is a fixed constant describing the maximal capacity. Similarly, constraints on the total diagnosis capacity can be described by

$$\int_{t_1}^{t_2} \int_0^\infty y(t - s)\mathcal{F}(t, s)h_d(s)\,ds\,dt < K.$$

To study the effect of limits to the tracing or testing capacity, we considered a state-dependent probability of tracing or diagnosis. In the case of limited tracing capacity, we modified $\varepsilon_c$ by

$$\hat{\varepsilon}_c(t) = \varepsilon_c \, \min\left\{1, \frac{K}{\text{individuals traced at time } t}\right\},$$

where $K$ denotes a maximal instantaneous capacity and the denominator is computed from (2.13). In the case of maximal diagnosis capacity, it was necessary to consider a time-dependent rate of diagnosis $\hat{h}_d(t, \tau)$ defined as

$$\hat{h}_d(t, \tau) = h_d(\tau) \, \min\left\{1, \frac{K}{\text{individuals diagnosed at time } t}\right\}.$$

# 3. Emerging epidemic and effective reproduction numbers

During an emerging epidemic, in the approximation $S(t) \approx N_0$, the force of infection is approximately exponential of the form

$$y(t) = y_0 e^{rt}, \tag{3.1}$$

where $r$ represents the Malthusian parameter, or real-time growth rate, of the epidemic. Under this assumption, the solutions of (2.11) are stationary in time, $h_c(t, \tau) = h_c(\tau)$, and satisfy

$$h_c(\tau) = \varepsilon_c \int_0^\infty \beta(s)e^{-rs}\mathcal{F}_d(\tau + s)\mathcal{F}_c(\tau + s)(h_d(\tau + s) + h_c(\tau + s))\,ds, \tag{3.2}$$

with $\mathcal{F}_c(\tau) = e^{-\int_0^\tau h_c(\sigma)d\sigma}$. Substituting (3.1) into (2.9) gives the Lotka–Euler-type equation

$$1 = \int_0^\infty \beta(s)e^{-rs}\mathcal{F}_d(s)\mathcal{F}_c(s)\,ds. \tag{3.3}$$

System (3.2) and (3.3) determines $r$ and $h_c$. It is interesting to note that, even in the setting of an emerging epidemic, the contact tracing process is described by a nonlinear equation (3.2).

We can define three reproduction numbers: the basic reproduction number $R_0$ with no diagnosis nor contact tracing, the reproduction number $R_d$ with diagnosis only, and the reproduction number $R_{d,c}$ when both diagnosis and contact tracing are in place. From (2.5), it is easy to verify that

$$R_0 = \int_0^\infty \beta(\tau)\,d\tau,$$

$$R_d = \int_0^\infty \beta(\tau)\mathcal{F}_d(\tau)\,d\tau$$

and

$$R_{d,c} = \int_0^\infty \beta(\tau)\mathcal{F}_d(\tau)\mathcal{F}_c(\tau)\,d\tau,$$

where $\mathcal{F}_c(\tau)$ is determined via (3.2).

The controllability of an infectious disease outbreak depends not only on the basic reproduction number $R_0$, but also on the proportion of transmission occurring before symptom onset, defined as

$$\theta_s := \left( \frac{\int_0^\infty \beta(\tau)\mathcal{F}_s(\tau)\,d\tau}{\int_0^\infty \beta(\tau)\,d\tau} \right),$$

where $\mathcal{F}_s(\tau)$ is the probability of not having developed symptoms by ASI $\tau$ [16]. In the presence of diagnosis only, or with additional contact tracing, the amount of transmission occurring before isolation is given by $\theta_d = R_d/R_0$ and $\theta_{d,c} = R_{d,c}/R_0$, respectively. It is therefore evident that, in the absence of vaccine or alternative intervention measures like physical distancing, an emerging disease which is by itself able to spread ($R_0 > 1$) can be brought under control only if the proportion of transmission occurring before isolation (either by diagnosis or contact tracing) is smaller than $1/R_0$.

A related important concept used to measure the effectiveness of contact tracing is the amount of onward transmission prevented by isolation [10,11]: this amounts to $1 - \theta_d$ for diagnosis only, and to $1 - \theta_{d,c}$ for diagnosis and contact tracing combined. In particular, the transmission prevented by contact tracing only (in addition to diagnosis) is $\theta_d - \theta_{d,c}$.

# 4. Applications to COVID-19

For the numerical simulations in this section, we used parameters specific to the COVID-19 pandemic, summarized in table 1 and electronic supplementary material, figure A.1. The probability density of the time to diagnosis is obtained by shifting the density of the incubation period to the right by a fixed amount, corresponding to the diagnosis delay; from the probability density and the corresponding cumulative distribution, the rate $h_d$ is computed from (2.2).

## 4.1. Impact of diagnosis and contact tracing on the control of the epidemic

Using the linear model (3.2) and (3.3), we investigated the role of diagnosis and contact tracing on the control of an epidemic. Figure 2 shows which combinations of diagnosis and contact tracing policies, in terms of diagnosis delay and coverage and contact tracing coverage, can attain control of the epidemic when the transmissibility is set at $R_0 = 2.5$ and $R_0 = 1.5$. These values were inspired by COVID-19 before and after the implementation of physical distancing. Although $R_0 = 2.5$ is at the low end of the spectrum of values estimated in the literature (e.g. [26]), it is useful to illustrate the challenge in controlling the epidemic through diagnosis and contact tracing alone.

Diagnosis alone is not sufficient to prevent the spread of the epidemic when $R_0 = 2.5$ (figure 2a). The situation improves for $R_0 = 1.5$, where diagnosis can control the epidemic if it is extremely efficient in terms of coverage and speed: diagnosis of 85% of cases (corresponding to the totality of clinical cases according to [23]) allows control only with delays of about 1 day or less from symptom onset. We remark, however, that this result is very sensitive to underlying assumptions about infectivity profile and incubation time distribution. The additional contact tracing process substantially improves the chances of controlling the epidemic under both scenarios, but with no physical distancing in place ($R_0 = 2.5$) requires a fast diagnosis in less than 2.5 days even when the coverage is perfect, or a tracing coverage of at least 60% even if diagnosis is immediate at symptoms. See also electronic supplementary material, figure A.2 for a further illustration of the substantial effect of diagnosis delays on $R_{d,c}$. Note also that, while reducing the effective reproduction number, contact tracing also shifts the generation time distribution towards shorter times (electronic supplementary material, figure A.3), thus also affecting the relationship between $R_{d,c}$ and the real-time growth rate.

## 4.2. Contact tracing efficacy in relation to tracing coverage and window

We investigated the effect of the tracing coverage (fraction of contacts effectively reached) and the tracing window (how many days back in the history of the index case contacts are considered for tracing). The influence of these aspects strongly depends on the amount of pre-symptomatic transmission and the infectiousness relative to symptoms [22,27]. For fixed infectiousness and incubation period distributions as in table 1, figure 3 shows the effect on $R_{d,c}$ and on the onward transmission prevented by contact tracing. From figure 3a, it is clear that enlarging the tracing window from 2 days up to 5 days substantially improves the controllability of the epidemic. However, figure 3b shows that increasing the coverage may have a more substantial impact on the amount of prevented

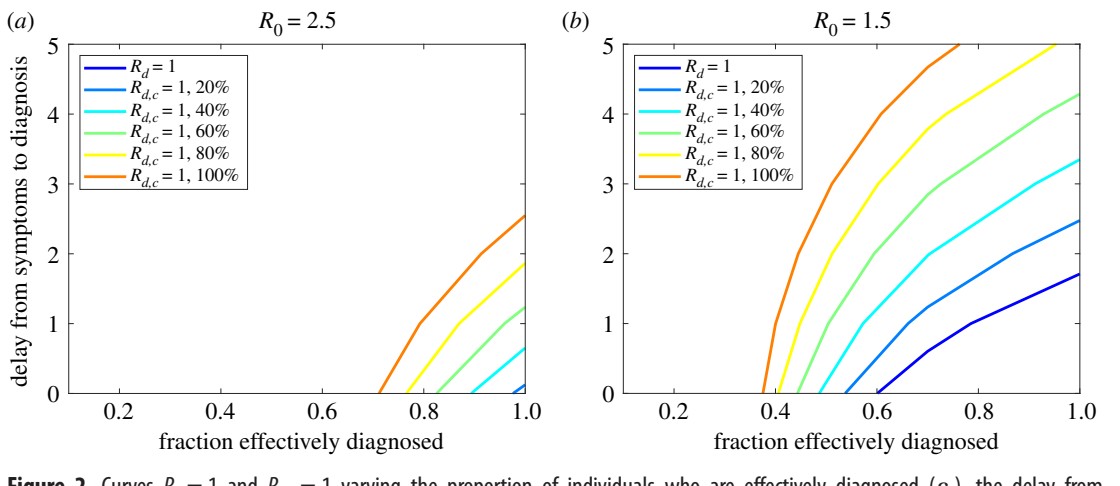

**Figure 2.** Curves $R_d = 1$ and $R_{d,c} = 1$ varying the proportion of individuals who are effectively diagnosed ($\varepsilon_d$), the delay from symptom onset to diagnosis, and the proportion of contacts who are effectively traced ($\varepsilon_c$, specified in the legend). For the underlying epidemic, we fixed $R_0 = 2.5$ (a) and $R_0 = 1.5$ (b). In the former case, diagnosis alone is never sufficient to prevent the spread of the epidemic.

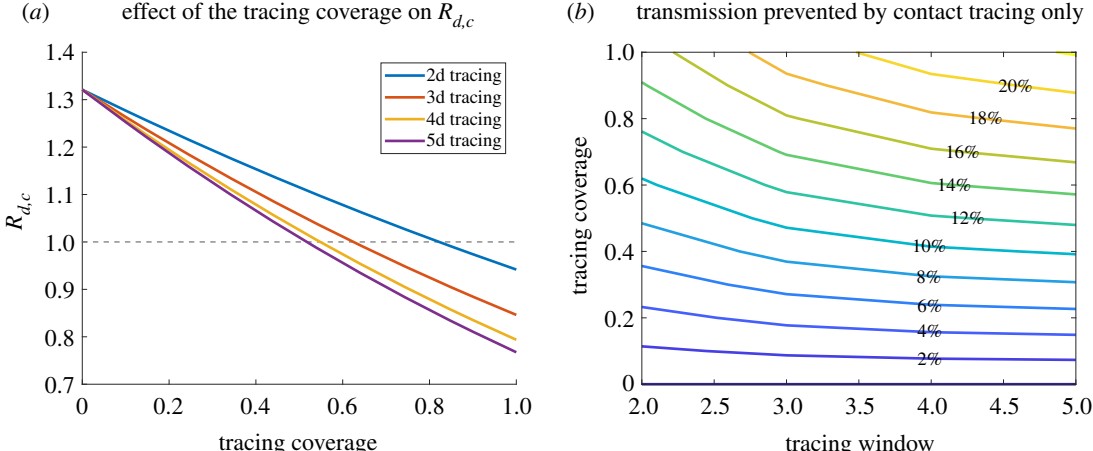

**Figure 3.** Effect of the tracing coverage ($\varepsilon_c$) and the length of the tracing window on $R_{d,c}$ (a) and on the fraction of prevented onward transmission in addition to diagnosis (b). We fixed $R_0 = 2.5$ and assumed no delay in diagnosis.

**Table 1.** Parameter values for COVID-19. Note that the infectiousness profile and the incubation period are independent of each other. With the specified parameters, approximately 45% of the transmission occurs prior to symptoms, consistently with [22,23]. The probability density of the time to diagnosis is obtained by shifting the density of the incubation period to the right by a fixed amount, corresponding to the diagnosis delay, with a 2-day delay in diagnosis corresponding to approximately 75% of the transmission occurring prior to diagnosis.

| model parameter | value | reference |
|---|---|---|
| incubation time | $\Gamma\ (\mu = 4.84, \ \sigma = 2.79)$ | [24] |
| infectiousness | $\Gamma\ (\mu = 5, \ \sigma = 1.9)$ | [25] |
| proportion of asymptomatic cases | 15% | [23] |
| maximal contact tracing window | 2–5 days | assumed |
| maximal time to diagnosis | 20 days | assumed |
| maximal duration of infectiousness | 20 days | assumed |

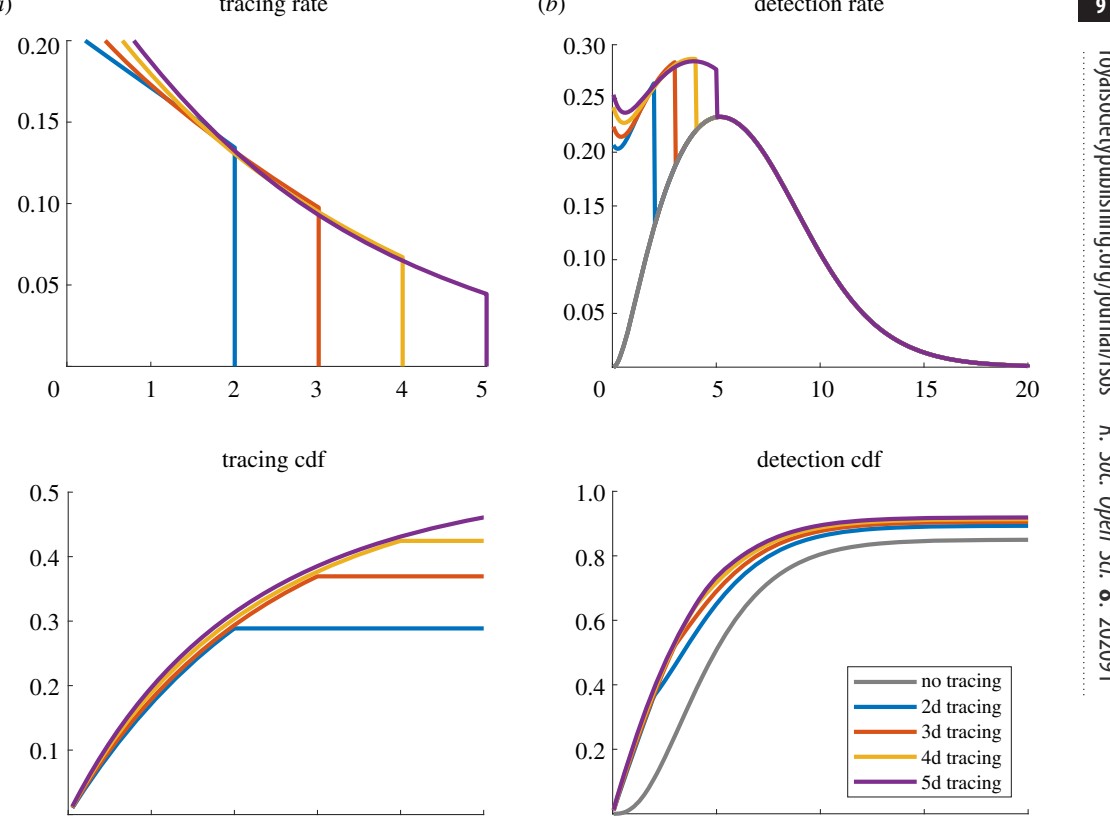

**Figure 4.** (*a*) Computed $h_c$ and cumulative distribution function of the time to isolation for contact tracing, for different lengths of the tracing window. The cumulative distribution function is computed as $1 - \mathcal{F}_c$ from (2.3). (*b*) The same quantities (rate and cumulative distribution function) for the overall detection from diagnosis and contact tracing combined. The parameters are taken as in figure 3, with $\varepsilon_c = 1$.

transmission. The impact on $R_{d,c}$, the real-time growth rate and the doubling time is shown in electronic supplementary material, figure A.4. Figure 4*a* shows the computed profiles of $h_c$ for $\varepsilon_c = 1$ and different lengths of the tracing window, and the cumulative distribution function of the time to detection for contact tracing, computed as $1 - \mathcal{F}_c$ from (2.3). Figure 4*b* shows the combination of tracing and diagnosis in terms of individual detection rate and cumulative distribution function of time to detection (via diagnosis or tracing). Similar analyses, but with $R_0 = 1.5$ and a 2-day diagnosis delay, are presented in electronic supplementary material, figures A.5–A.7.

## 4.3. Impact of contact tracing interruption under constraints on testing capacity

The interruption of contact tracing corresponds, in figures 2 and 3, to set the tracing coverage to zero, hence causing naturally an increase in the effective reproduction number and in the real-time growth rate.

By simulating the epidemic outbreak using the nonlinear model (2.9)–(2.11), including maximal capacity either in testing or tracing, we investigated the possible effects of an interruption of contact tracing with different duration and underlying epidemic growth; see figure 5. The upper panels show the case of an epidemic in which optimal contact tracing is not sufficient to control the epidemic. Interruption of contact tracing causes an increase in growth rate and breaching of testing/tracing capacity. The middle panels show a case in which diagnosis and contact tracing maintain control of the epidemic both before and after the temporary interruption of contact tracing: since tracing capacity is never reached, the reintroduction of contact tracing eventually restores the original growth rate and allows the control of the epidemic. When the constraints are on diagnosis (right panels) the effect is delayed since diagnosis capacity is breached, but contact tracing still brings the growth rate to original levels (and diagnosis again under capacity). The lower panels illustrate a qualitative worst-case scenario in which the epidemic

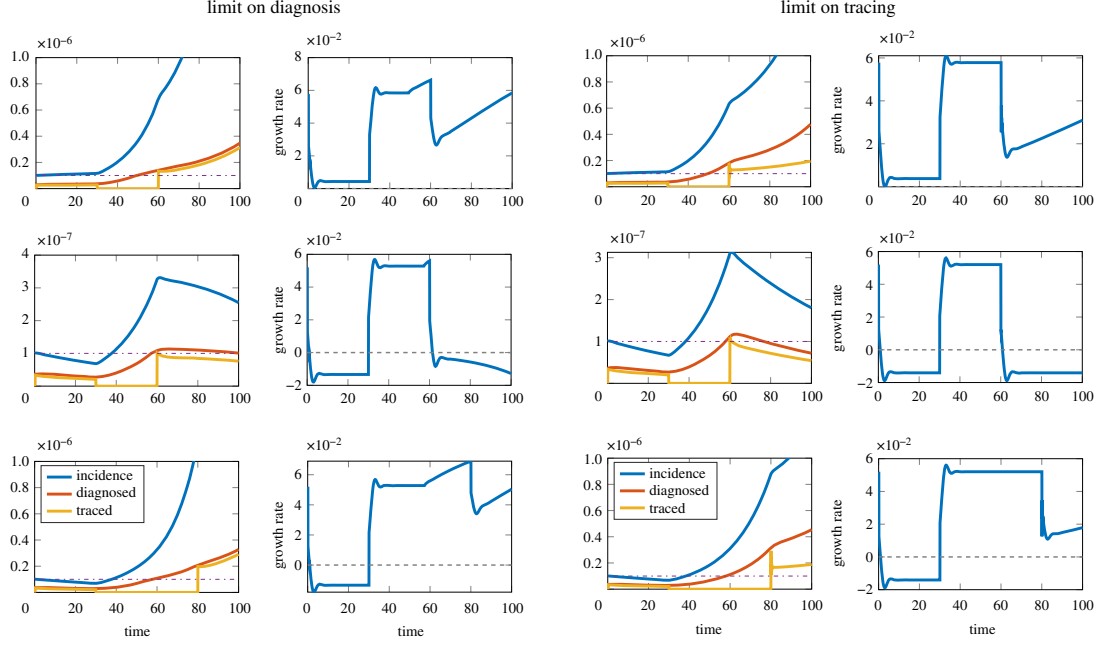

**Figure 5.** Possible effects of a short-term interruption of contact tracing in the presence of maximal capacity, for $R_0 = 1.5$ and a 2-day diagnosis delay. On the left, capacity is on tracing (max $10^{-7}$ total tracing rate); on the right, capacity is on diagnosis (max $10^{-7}$ total diagnosis rate). First row: 50% of infected individuals are diagnosed ($\varepsilon_d = 0.5$), the epidemic grows in the presence of contact tracing; the interruption causes a temporary increase in the real-time growth rate and the breaching of capacity (which would have occurred eventually anyway), which leads to increasing growth rate. Second row: 60% of infected individuals are diagnosed ($\varepsilon_d = 0.6$), the epidemic is controlled in the presence of contact tracing; interruption of contact tracing increases the value of the growth rate, but reintroducing contact tracing after $T = 30$ days brings the growth rate to the original level; this is delayed in the case of capacity on diagnosis (right), as the capacity is breached during the contact tracing interruption. Third row: 60% of infected individuals are diagnosed ($\varepsilon_d = 0.6$), the epidemic is controlled in the presence of contact tracing; the interruption of contact tracing for $T = 50$ days causes the breaching of capacity, so that when contact tracing is resumed it is not sufficient to control the epidemic, and an outbreak is observed. The initial condition for the infected individuals was taken as $10^{-7} e^{r_d t}$, where $r_d$ is the growth rate due to diagnosis only. Oscillations in the observed growth rate are an effect of the numerical method.

is initially controlled, but interruption of contact tracing is long enough to make the loss of control irreversible: even after the reintroduction of contact tracing the growth rate will remain positive, and will increase due to the increasing prevalence. This analysis is particularly relevant for regions with lower prevalence and growth rates (for example, some rural areas), where the disease may be controllable by contact tracing. We have included the computed profiles of $h_c$ from the nonlinear model in electronic supplementary material, figure A.8, before and after each contact tracing interruption.

## 4.4. Impact of physical distancing versus contact tracing

The effectiveness of a contact tracing policy depends not only on the tracing efforts, but also on the epidemiological properties of the disease. The same contact tracing policy (in terms of tracing coverage and length of the tracing window) can control an epidemic with low transmissibility, but not an epidemic with high transmissibility. Figure 6 shows the relation of $R_{d,c}$ with the contact tracing coverage ($\varepsilon_c$) and the unconstrained reproduction number $R_0$. This relation has important public health implications, especially for planning economic investments. Intensifying contact tracing efforts (e.g. by employing larger number of tracers or developing and deploying digital tracing systems), translates into a shift upward in the plane in figure 6, whereas reducing overall contact (e.g. by intensifying physical distancing measures), translates into a shift to the left. Whereas contact tracing efforts may be sufficient to prevent the epidemic from taking off in some regions, strict physical distancing measures (e.g. mass quarantine) may be eventually unavoidable in regions with higher transmissibility. Considering limitations to the minimum per capita contact rate or to the maximal contact tracing effort (e.g. due to adherence of individuals to isolation or to disclose contacts [28]) is fundamental for planning interventions.

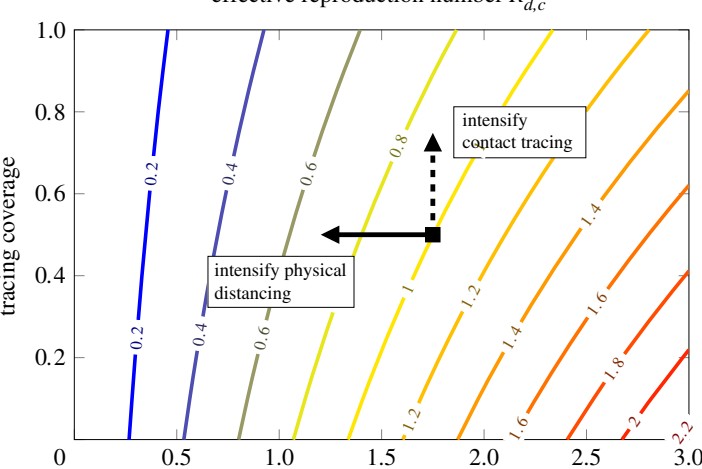

**Figure 6.** Level curves of $R_{d,c}$ in the $(R_0, \varepsilon_c)$-plane. Reducing $R_0$ can be obtained by reducing the contact rate via non-pharmaceutical interventions like physical distancing or mass quarantine. Increasing $\varepsilon_c$ can be obtained by intensifying contact tracing. Note that, in the presence of testing/tracing maximal capacity, the detrimental effect of reducing $\varepsilon_c$ is enhanced.

## 5. Discussion and conclusion

We formulated a deterministic model for disease transmission dynamics including both diagnosis, either from symptom onset and hospitalization or from any public health screening/monitoring programme, and contact tracing. As many important indicators of public health relevance can be computed directly and efficiently from the model variables, the model can represent a valid support to inform public response to emerging epidemics, as it easily allows investigation of scenarios for the effects of several non-pharmaceutical interventions. We used the model to investigate several mechanisms of public health relevance using parameters characteristic of the COVID-19 pandemic.

A first important aspect is the large impact of detection delays on the control of the epidemic. For the COVID-19 pandemic, many researchers noticed early on that, due to the large potential pre-symptomatic transmission, only high contact tracing coverage coupled with very short delays would be effective to stop the spread, thus supporting the implementation of digital contact tracing as a tool to minimize the isolation delays [8,10–13,25]. Our findings show that, even with extremely quick diagnosis and tracing, the coverage level required to reach control in the case of COVID-19 suggests that digital contact tracing is probably insufficient, unless the current low app usage observed in different countries is significantly improved [29]. A significant reduction in the underlying transmission seems anyhow necessary for any realistic contact tracing or diagnosis programme to be successful in gaining control, either obtained through physical distancing or by vaccination programmes.

In the presence of pre-symptomatic transmission, as in the case of COVID-19, the length of the tracing window, i.e. the number of days back from the detection of the index case during which contacts are traced, is fundamental for the effectiveness of contact tracing. For COVID-19, it was suggested that tracing 2 days before detection may not be sufficient to cover enough pre-symptomatic transmission [22,23,27]. We investigated different choices of contact tracing windows from 2 to 5 days and found that, for $R_0 = 2.5$ and the other parameters assumed here, if less than 80% of contacts are traced, tracing only 2 days backward is not sufficient to control the epidemic. A tracing window of 5 days makes control achievable but only if the tracing coverage is no less than 50%.

In the presence of limited testing or tracing capacity, diagnosis or tracing efforts may need to be redistributed among regions with high or low prevalence, with some regions being prioritized over others. We showed how these decisions must take into account several aspects related to the underlying transmission and the overall testing/tracing capacity of the system. A short-term interruption of contact tracing in a region where the epidemic is apparently under control may have dramatic impact on the outbreak, as, with limited capacity, the loss of control may be irreversible: reintroducing contact tracing when the tracing capacity has been exceeded does not recover the original decline. A careful preliminary evaluation is fundamental especially in response to COVID-19 since, due to possibly significant delays between infection and detection, the effects of the interruption of contact tracing will be visible only after several days [30].

To our knowledge, this is the first deterministic model for contact tracing formulated in terms of a mechanistic description of the individual-level processes (disease transmission, diagnosis and contact tracing). A similar construction is used for instance in [6–8] in the formulation of a stochastic branching process. We remark, however, that, although the conclusions are in line with the ones discussed here for the linear approximation to the initial epidemic phase, the generation perspective of the branching process does not allow to capture the full nonlinear epidemic dynamics. Most of the current scientific literature addressing contact tracing relies on stochastic agent-based models. Compared with the latter, deterministic models have several advantages: first, they are computationally more efficient and faster to simulate, which is crucial when simulations must be repeated many times, for instance in model fitting and parameter estimation; second, they often allow to obtain analytically tractable and more transparent relations between structural model assumptions and parameter values, and the conclusions.

The model has several limitations that could be addressed in future work. First, it is based on the assumption of homogeneous mixing, so clustered contact tracing, for instance focused on households or realistic social networks [7,13,31–33], is not captured in this framework. Some level of heterogeneity could be incorporated by considering a vector-valued incidence and contact mixing matrices. Second, to simplify the presentation of the model and to obtain preliminary insights, we have here assumed instantaneous contact tracing. More realistic tracing delays could be incorporated. Third, in the current model, the infectivity profile is assumed to be independent of the incubation time. Given the efficacy of contact tracing is strongly affected by the amount of pre-symptomatic transmission, an important extension would consist in incorporating the correlation between individual infectiousness and onset of symptoms. Fourth, the model only tracks infected individuals, ignoring contacts that did not lead to an infection. The advantage is that our construction does not require specifying the contact rate in the overall population, but only the infectivity profile of infected individuals. However, by making assumptions about the overall contact rate, we could further describe the tracing and possible quarantine of contacted individuals that escaped infection. Quarantine of susceptible individuals is a fundamental aspect for socio-economic evaluations of the impact of a contact tracing policy in terms of resources invested in the contact tracing effort and disruptions to healthy individuals [11]. Finally, although preliminary tests we carried out appear promising, more work is needed in investigating the comparison between the output of the simulation of the nonlinear deterministic model and the average prediction of an agent-based model describing the same processes in a homogeneously mixing population.

Data accessibility. Data and relevant code for this research work are stored in GitHub: https://github.com/scarabel/Contact-Tracing and have been archived within the Zenodo repository: https://doi.org/10.5281/zenodo.4626185.

Authors' contributions. F.S., L.P. and J.W. designed the study; F.S. developed the model, performed the numerical simulations and wrote the first draft of the manuscript; all authors contributed to the revision and interpretation of the results and to the final version of the manuscript.

Competing interests. We declare we have no competing interests.

Funding. This research of F.S. and J.W. has been funded by the Canadian Institute of Health Research (CIHR) 2019 Novel Coronavirus (COVID-19) rapid research program. F.S. is a member of the INdAM Research group GNCS. L.P. acknowledges funding from the Wellcome Trust and the Royal Society (grant no. 202562/Z/16/Z), and by the UKRI through the JUNIPER modelling consortium (grant no. MR/V038613/1). J.W. is a member of the Ontario COVID-19 Modelling Consensus Table, sponsored by the Ontario Ministry of Health, Ontario Health, and Public Health Ontario.

Acknowledgements. The authors acknowledge two anonymous reviewers for helpful comments that improved the readability of the manuscript. F.S. is grateful to Eugenia Franco for useful discussions on some mathematical aspects of the model.

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
