## [Peer Review File · Royal Society Open Science]

Review History

RSOS-202091.R0 (Original submission)

Review form: Reviewer 1

Is the manuscript scientifically sound in its present form?

Yes

Are the interpretations and conclusions justified by the results?

Yes

Is the language acceptable?

Yes

Do you have any ethical concerns with this paper?

No

Have you any concerns about statistical analyses in this paper?

No

Recommendation?

Accept with minor revision (please list in comments)

Comments to the Author(s)

See the attached PDF (Appendix A).

Review form: Reviewer 2

Is the manuscript scientifically sound in its present form?

Yes

Are the interpretations and conclusions justified by the results?

Yes

Is the language acceptable?

Yes

Do you have any ethical concerns with this paper?

No

Have you any concerns about statistical analyses in this paper?

No

Recommendation?

Accept with minor revision (please list in comments)

Comments to the Author(s)

In this paper, the authors formulated a renewal equation model to describe the complex process of contact tracing and investigated its effects in the control of COVID-19. This paper is very well-written and the formulation of the contact tracing process is clear and concise. I would highly recommend this paper for publication, and I have some minor comments and questions listed below.

1. The authors should consider re-organizing the terms on the right-hand side of the derivation for $h_c(t, \tau) dt$ on page 9 and between (2.7) and (2.8).
2. The simulations of the model were conducted either for the system (2.6)-(2.8) or (3.2)-(3.3), where the hazard rate of detection h_d was prescribed. It would be helpful if the authors can point out how the formulation of h_d is related to the model parameters illustrated in Figure 1A and Table 1.
3. What would the shape of h_c look like in terms of a solution to the model system?
4. The authors stated that they have cross-checked their model with an ABM model on page 20, it would be helpful if they can provide detailed references/links about the ABM model.

Decision letter (RSOS-202091.R0)

This year has been very difficult for everyone, and we want to take the opportunity to thank you for your continued support in 2020.

The Royal Society Open Science editorial office will be closed from the evening of Friday 18 December 2020 until Monday 4 January 2021. We will not be responding during this time. If you

have received a deadline within this time period, please contact us as soon as possible to allow us to extend the deadline. If you receive any automated messages during this time asking you to meet a deadline, we offer apologies and invite you to respond after the festive period or during normal working hours.

With our best for a peaceful festive period and New Year, and we look forward to working with you in 2021.

Dear Dr Scarabel

On behalf of the Editors, we are pleased to inform you that your Manuscript RSOS-202091 "A renewal equation model to assess roles and limitations of contact tracing for disease outbreak control" has been accepted for publication in Royal Society Open Science subject to minor revision in accordance with the referees' reports. Please find the referees' comments along with any feedback from the Editors below my signature.

Please submit your revised manuscript and required files (see below) no later than 7 days from today's (ie 14-Dec-2020) date. Note: the ScholarOne system will 'lock' if submission of the revision is attempted 7 or more days after the deadline. If you do not think you will be able to meet this deadline please contact the editorial office immediately.

on behalf of Dr Pierre Magal (Associate Editor) and Glenn Webb (Subject Editor)
openscience@royalsociety.org

Reviewer comments to Author:

Reviewer: 1

Comments to the Author(s)

See the attached PDF

Reviewer: 2

Comments to the Author(s)

In this paper, the authors formulated a renewal equation model to describe the complex process of contact tracing and investigated its effects in the control of COVID-19. This paper is very well-

written and the formulation of the contact tracing process is clear and concise. I would highly recommend this paper for publication, and I have some minor comments and questions listed below.

1. The authors should consider re-organizing the terms on the right-hand side of the derivation for $h_c(t, \tau) dt$ on page 9 and between (2.7) and (2.8).
2. The simulations of the model were conducted either for the system (2.6)-(2.8) or (3.2)-(3.3), where the hazard rate of detection h_d was prescribed. It would be helpful if the authors can point out how the formulation of h_d is related to the model parameters illustrated in Figure 1A and Table 1.
3. What would the shape of h_c look like in terms of a solution to the model system?
4. The authors stated that they have cross-checked their model with an ABM model on page 20, it would be helpful if they can provide detailed references/links about the ABM model.

===PREPARING YOUR MANUSCRIPT===

===PREPARING YOUR REVISION IN SCHOLARONE===

Author's Response to Decision Letter for (RSOS-202091.R0)

See Appendix B.

Decision letter (RSOS-202091.R1)

Dear Dr Scarabel,

I am pleased to inform you that your manuscript entitled "A renewal equation model to assess roles and limitations of contact tracing for disease outbreak control" is now accepted for publication in Royal Society Open Science. Please accept our apologies for the delay.

At this stage, we ask that you please archive your GitHub code within the Zenodo repository: <https://guides.github.com/activities/citable-code/>. By doing this, a formal, citable DOI will be associated with your data record, and an open license (CC-BY preferred) can be applied to your data. We would then ask that you please update your data availability statement to read as:

"Data and relevant code for this research work are stored in GitHub: [GitHub URL here] and have been archived within the Zenodo repository: <https://doi.org/zenodo.....> [ref number].

If you have not already done so, please remember to make any other data sets or code libraries 'live' prior to publication, and update any links as needed when you receive a proof to check - for instance, from a private 'for review' URL to a publicly accessible 'for publication' URL. It is good practice to also add data sets, code and other digital materials to your reference list.

COVID-19 rapid publication process:

We are taking steps to expedite the publication of research relevant to the pandemic. If you wish, you can opt to have your paper published as soon as it is ready, rather than waiting for it to be published the scheduled Wednesday.

This means your paper will not be included in the weekly media round-up which the Society sends to journalists ahead of publication. However, it will still appear in the COVID-19 Publishing Collection which journalists will be directed to each week (<https://royalsocietypublishing.org/topic/special-collections/novel-coronavirus-outbreak>).

If you wish to have your paper considered for immediate publication, or to discuss further, please notify openscience_proofs@royalsociety.org and press@royalsociety.org when you respond to this email.

on behalf of Dr Pierre Magal (Associate Editor) and Glenn Webb (Subject Editor)
openscience@royalsociety.org

Appendix A

Report on Manuscript “A renewal equation model to assess roles and limitations of contact tracing for disease outbreak control”

The manuscript authored by Francesca Scarabel and collaborators is concerned with a deterministic model of contact-tracing in the context of an epidemic outbreak. More precisely, the authors study an epidemic model with an age since infection (ASI) structure, and introduce contact-tracing as a time-dependent factor in the transmission chain. They draw a link with various relevant quantities including the basic reproductive number R_0 and the proportion of transmissions occurring before symptom onset. The article is supplemented with numerical applications with parameter values which are consistent with the COVID-19 outbreak.

Overall I think that the article is remarkably well-written and very clear in the exposition. It provides a timely and highly desirable study of contact-tracing as a control measure and even provides quantitative cues on the intensity that is required to control the epidemic and possible consequences of a deviation in the contact-tracing strategy. I only have two remarks, listed below, that I think would help improve the readability of the paper. Otherwise I strongly recommend this article for publication in the Royal Society Open Science.

1. The term “survival probability” used on line 36 of page 6 and line 11 of page 7, though widely accepted by the mathematical community, may be misinterpreted in the context of the paper. In my opinion it would be better to avoid this term altogether, unless a non-ambiguous synonym can be found. Similarly, the term “hazard” is somewhat confusing.
2. The evolution equation for the quantities $S(t)$ and $I(t, \tau)$ (the density of infected) are never explicitly written. I believe that it would help readers less accustomed to the ASI structure to recall the complete system somewhere in the manuscript. It would also help to precise the links between the force of infection $\Lambda(t)$ and $I(t, \tau)$. This could possibly be included as a supplementary material and referenced in the main text.

Appendix B

Response to editor and reviewers

We are grateful to the reviewers for their kind words and for their constructive comments. We have tried to address every issue raised by the reviewers and we include point-by-point responses below.

During the revision we have also taken the opportunity to improve some sentences in the text and to add some further references of which we became aware only after submission, and we considered relevant for the work. All changes from the previous version are highlighted in blue color. No change was made to the mathematical construction and the results.

Reviewer 1

The manuscript authored by Francesca Scarabel and collaborators is concerned with a deterministic model of contact-tracing in the context of an epidemic outbreak. More precisely, the authors study an epidemic model with an age since infection (ASI) structure, and introduce contact-tracing as a time-dependent factor in the transmission chain. They draw a link with various relevant quantities including the basic reproductive number R_0 and the proportion of transmissions occurring before symptom onset. The article is supplemented with numerical applications with parameter values which are consistent with the COVID-19 outbreak.

Overall I think that the article is remarkably well-written and very clear in the exposition. It provides a timely and highly desirable study of contact-tracing as a control measure and even provides quantitative cues on the intensity that is required to control the epidemic and possible consequences of a deviation in the contact-tracing strategy. I only have two remarks, listed below, that I think would help improve the readability of the paper. Otherwise I strongly recommend this article for publication in the Royal Society Open Science.

- The term “survival probability” used on line 36 of page 6 and line 11 of page 7, though widely accepted by the mathematical community, may be misinterpreted in the context of the paper. In my opinion it would be better to avoid this term altogether, unless a non-ambiguous synonym can be found. Similarly, the term “hazard” is somewhat confusing.

We agree that this terminology could be confusing in this context and we have happily accepted the suggestion of the reviewer by removing the terms “survival” and “hazard” altogether.

- The evolution equation for the quantities $S(t)$ and $I(t,\tau)$ (the density of infected) are never explicitly written. I believe that it would help readers less accustomed to the ASI structure to recall the complete system somewhere in the manuscript. It would also help to precise the links between the force of infection $\Lambda(t)$ and $I(t,\tau)$. This could possibly be included as a supplementary material and referenced in the main text.

We have added a specification about $I(t,\tau)$ in the main text (section “A renewal equation for the incidence”) together with the equation for $S(t)$, and we have recalled the equations in the section “Interpretation of the model”. We hope these equations give enough information also to readers that are less familiar with the ASI formulation.

Reviewer 2

In this paper, the authors formulated a renewal equation model to describe the complex process of contact tracing and investigated its effects in the control of COVID-19. This paper is very well-written and the formulation of the contact tracing process is clear and concise. I would highly recommend this paper for publication, and I have some minor comments and questions listed below.

- The authors should consider re-organizing the terms on the right-hand side of the derivation for $h_c(t,\tau) dt$ on page 9 and between (2.7) and (2.8).

We are not sure we have interpreted correctly the reviewer's suggestion. We have corrected the definition of the probability in (2.10) as $psi ds$, and we have included the following specification about the right-hand side of the derivation for $h_c(t,\tau) dt$: "Note that the integral at the right-hand side is computed with respect to s ". We have left the explanation in words inside the equations, as we believe it makes the derivation more transparent.

We hope these additions improve the readability of the equations, but we remain open to further suggestions.

- The simulations of the model were conducted either for the system (2.6)-(2.8) or (3.2)-(3.3), where the hazard rate of detection h_d was prescribed. It would be helpful if the authors can point out how the formulation of h_d is related to the model parameters illustrated in Figure 1A and Table 1.

We thank the reviewer for the helpful suggestion. We have now added this specification in equation (2.2) in the new version and recalled it at the beginning of Section 4.

- What would the shape of h_c look like in terms of a solution to the model system?

This is a very interesting question and we were happy to include some profiles in the manuscript. They are now included in Figure 4, for different tracing windows in the linear model, and in the Supplementary Figures A.7 and A.8, the latter obtained with the nonlinear model.

- The authors stated that they have cross-checked their model with an ABM model on page 20, it would be helpful if they can provide detailed references/links about the ABM model.

We agree that this point would require further specification. However, we feel that the work in that direction is still not mature enough for publication, hence we have slightly changed the paragraph by clarifying that we plan to study more rigorously the comparison with the ABM in future work. Precisely we changed the paragraph to:

"Finally, although preliminary tests we carried out appear promising, more work is needed in investigating the comparison between the output of the simulation of the nonlinear deterministic model and the average prediction of an agent-based model describing the same processes in a homogeneously mixing population."